# The Impact of Coilin Nonsynonymous SNP Variants E121K and V145I on Cell Growth and Cajal Body Formation: The First Characterization

**DOI:** 10.3390/genes11080895

**Published:** 2020-08-05

**Authors:** Yue Yao, Heng Wee Tan, Zhan-Ling Liang, Gao-Qi Wu, Yan-Ming Xu, Andy T. Y. Lau

**Affiliations:** 1Laboratory of Cancer Biology and Epigenetics, Shantou University Medical College, Shantou 515041, Guangdong, China; 15yyao1@stu.edu.cn (Y.Y.); hwtan@stu.edu.cn (H.W.T.); 16zlliang@stu.edu.cn (Z.-L.L.); 17gqwu@stu.edu.cn (G.-Q.W.); 2Department of Cell Biology and Genetics, Shantou University Medical College, Shantou 515041, Guangdong, China

**Keywords:** *COIL*, single nucleotide polymorphism, mutation, SMN, cancer risk

## Abstract

Coilin is the main component of Cajal body (CB), a membraneless organelle that is involved in the biogenesis of ribonucleoproteins and telomerase, cell cycle, and cell growth. The disruption of CBs is linked to neurodegenerative diseases and potentially cancers. The coilin gene (*COIL*) contains two nonsynonymous SNPs: rs116022828 (E121K) and rs61731978 (V145I). Here, we investigated for the first time the functional impacts of these coilin SNPs on CB formation, coilin subcellular localization, microtubule formation, cell growth, and coilin expression and protein structure. We revealed that both E121K and V145I mutants could disrupt CB formation and result in various patterns of subcellular localization with survival motor neuron protein. Noteworthy, many of the E121K cells showed nucleolar coilin accumulation. The microtubule regrowth and cell cycle assays indicated that the E121K cells appeared to be trapped in the S and G2/M phases of cell cycle, resulting in reduced cell proliferation. In silico protein structure prediction suggested that the E121K mutation caused greater destabilization on the coilin structure than the V145I mutation. Additionally, clinical bioinformatic analysis indicated that coilin expression levels could be a risk factor for cancer, depending on the cancer types and races.

## 1. Introduction

Cajal body (CB), a membraneless subnuclear organelle, is essential for the assembly and modification of small nuclear ribonucleoproteins (snRNPs), small nucleolar ribonucleoproteins (snoRNPs), and telomerase [1]. CBs or CB-like structures can be found in many living organisms, including human, mammalian, plant, yeast, amphibian, and insect [2,3]. However, CBs are not necessarily observable in all cell types, but more commonly in those that are transcriptionally active and/or with high splicing demands, such as neuronal and cancerous cells [4,5]. For cells that harbor CBs, the number and size of CBs could change according to their proliferative and metabolic status, indicating the involvement of CBs in cell growth [6,7,8].

The CB consists of several important components, including coilin and survival motor neuron (SMN) protein [9]. The human coilin is encoded by the gene *COIL* (HGNC: 2184), and it is considered as the architectural element and marker protein of the CBs. Structural studies of coilin have revealed that this protein contains several conserved domains, such as a self-interacting domain [10], a cryptic nucleolar localization signal (NoLS) located between two nuclear localization sequences (NLSs) [11], a tri-RG motif [12], and a tudor domain [13].

CB dysfunction has been linked to human inherited neurodegenerative diseases and cancers. However, the direct connections between coilin and human diseases have remained unclear. So far, the main culprits that responsible for CB-related diseases have always been identified as the proteins that localized to CBs, but rarely as coilin itself [14,15]. Spinal muscular atrophy (SMA) is a typical example of CB-related disease that is mainly caused by the mutations in the *SMN1* gene [16]. SMA is a notorious autosomal recessive motor neuron disease and a leading genetic cause of early childhood mortality with high carrier frequency—about one in 30 to 40, resulting in the occurrence of approximately 1:6000 live births [16]. In SMA patients, the SMN proteins failed to accumulate in CBs to form functional CBs, which revealed the importance of co-localization between coilin and SMN in the CBs [8]. Moreover, even though many studies have shown that CBs exist in a range of cancer cell lines [2,17], the relationships between CB, coilin expression, and cancer have remained largely elusive.

Single nucleotide polymorphism (SNP) refers to the genetic variation among people that occurred in at least 1% of the population [18]. In recent years, large-scale sequencing projects to determine the SNPs of various human populations throughout the globe have been carried out [19,20,21]. Evidence has shown that SNP is one of the potential factors that determine the risk of hundreds of diseases, including cancers [22,23,24]. Among the SNP types, the nonsynonymous (missense) SNPs have the highest probability of affecting protein functions, changing cell phenotypes, and resulting in diseases [25].

According to Single Nucleotide Polymorphism Database (dbSNP; https://www.ncbi.nlm.nih.gov/snp/), the human *COIL* gene contains two nonsynonymous SNPs: rs116022828 (E121K, c.361G>A, p.Glu121Lys) and rs61731978 (V145I, c.433G>A, p.Val145Ile). Both of these SNPs are located between the two NLSs of coilin, where a cryptic NoLS is positioned. The NoLS and NLS are essential for the subcellular localization of coilin, but, to date, the effects of coilin SNP variants on CB formation and other cell phenotypes such as cell growth, have not been investigated. The main objective of this study is to assess the roles of coilin nonsynonymous SNPs in CB formation and cell growth-related phenotypes. Here, we constructed HeLa cell lines that expressed wild-type (WT) or mutant forms (E121K and V145I) of coilin, and investigated the functional impacts of these SNPs on CB formation, coilin subcellular localization, microtubule formation, cell cycle, cell proliferation, and coilin expression and protein structure. Additionally, clinical bioinformatic analysis was also performed in order to evaluate the possible relationships between coilin and cancers.

## 2. Materials and Methods

### 2.1. Plasmid Construction

The human coilin coding sequence was amplified by PCR from cDNA of normal human bronchial epithelial cells (BEAS-2B) and inserted into pEGFP-C1 through XhoI and BamHI. Mutants were generated using the QuickChange site-directed mutagenesis kit (#210519, Agilent, Palo Alto, CA, USA). The primers used are listed in Appendix A. All of the constructs were verified by sequencing.

### 2.2. Cell Culture and Transfection

HeLa cells were cultured in MEM medium that was supplemented with 10% fetal bovine serum and 1% penicillin/streptomycin. Transient transfections of plasmid DNA were performed in antibiotic-free medium using PEI. Stable cell lines were obtained by 800 µg/mL G418 (Thermo Fisher Scientific, Frederick, MD, USA) selection medium. Representative clones were picked and used in this study.

### 2.3. Immunofluorescence Microscopy

The cells were grown on cover glass placed in six-well plates, and then fixed with 4% paraformaldehyde for 15 min. at room temperature upon analysis. Briefly, 0.1% (v/v) Triton X-100 was added for permeabilization and incubated on ice for 20 min. After blockage with 5% BSA in PBS for 1.5 h, the cells were subsequently incubated with SMN antibody (#12976, Cell Signaling Technology, Danvers, MA, USA) at 4 °C overnight and, the next day, incubated with secondary antibody Dylight^®^ 594 (red) (35510, Thermo Fisher Scientific, Waltham, MA, USA) for 1.5 h at room temperature. Nuclear DNA was stained with 1 µg/mL Hoechst 33258 (blue) (B1155, Sigma–Aldrich, Taufkirchen, Germany) at room temperature for 5 min. The images were captured using a LSM880 confocal laser scanning microscope (Zeiss, Oberkochen, Germany) at 400× magnification.

### 2.4. Microtubule Regrowth Assay

Cells growing on glass coverslips were treated with nocodazole (150 ng/mL) for 16 h at 37 °C. The cells were then fixed with 4% paraformaldehyde for 15 min. at room temperature 1 h after the removal of the nocodazole. The cells were subjected to immunofluorescence assay using a tubulin antibody (sc-5286, Santa Cruz Biotechnology, Santa Cruz, CA, USA) followed by Alexa Fluor Plus 555 secondary antibody (orange) (A32727, Thermo Fisher Scientific, Waltham, MA, USA).

### 2.5. Cell Cycle Analysis

The cells were seeded into 60 mm dishes and starved for 24 h, followed by collection at 0 h and at 12 h after being stimulated by 10% FBS. For cell collection, the cells were trypsinized and fixed with pre-chilled 70% ethanol. Then, cells were stained with propidium iodide/RNase staining solution (4087S, Cell Signaling Technology, Danvers, MA, USA) for 15 min. at room temperature and analyzed for cell cycle phases using BD Accuri™ C6 Flow Cytometer (BD Bioscience Co., Ltd., San Diego, CA, USA). Data generated were analyzed using FlowJo software (v7.6.1, TreeStar, Ashland, OR, USA).

### 2.6. Cell Proliferation Assay

Cell growth was determined by 3-(4,5-dimethylthiazol-2-yl)-5-(3-carboxymethoxyphenyl)-2-(4-sulfophenyl)-2H-tetrazolium (MTS) assay according to the manufacturer’s protocol (Promega, Madison, WA, USA) and as described previously [26]. Briefly, the cells were seeded in 96-well plates at 1000 cells per well, and MTS assay was performed in a 12 h interval for up to 60 h.

### 2.7. Protein Expression Examination

Immunoblotting was performed in order to examine the protein expression level of coilin. The proteins were separated by SDS-PAGE and then transferred to polyvinylidene difluoride membranes (Merck-Millipore, Darmstadt, Germany), as described previously [27]. Antibodies used were purchased from Santa Cruz Biotechnology: GFP (sc-9996) and coilin (sc-55594) and Sigma–Aldrich: β-actin (A5441). The bands were detected by chemiluminescence (Tanon5200, Tanon Science & Technology Co., Ltd., Shanghai, China).

### 2.8. Gene Expression Examination

Real-time quantitative PCR (RT-qPCR) was performed in order to examine the gene expression level of coilin using an ABI7500 Real-Time PCR System (Applied Biosystems, Foster City, CA, USA), and calculated using the comparative C_T_ method as described previously [26]. Primers used are listed in Appendix A. *ACTB* was amplified as an internal reference to normalize gene expression.

### 2.9. Protein Structure Prediction and Structural Bioinformatic Analysis

A hierarchical approach, the Iterative Threading ASSEmbly Refinement (I-TASSER) server, was used to predict the protein structure of the coilin WT and the two SNP variants (https://zhanglab.ccmb.med.umich.edu/I-TASSER) [28]. Predicted normalized B-factor and C-score based on the amino acid (AA) sequences that were sent to the server were generated. The predicted normalized B-factor is a value that indicates the stability of the residues in the protein structure: negative value means the residue is relatively more stable in the structure. The C-score, in which values can be ranging from −5 to +2, is a score to indicate the estimated global accuracy of the model. A C-score >−1.5 indicates a model of correct global topology. STRUM was used to predict the stability of a protein upon single-point mutation (https://zhanglab.ccmb.med.umich.edu/STRUM). ΔΔG values were generated for each mutation points, and a positive ΔΔG value indicates that the mutation induces protein stability, whereas a negative value causes destabilization [29].

### 2.10. Clinical Bioinformatic Analysis

Kaplan–Meier Plotter database (http://kmplot.com) was utilized to assess the correlation between coilin gene expression and survival of patients in 21 types of cancers [30]. The survival analysis of each cancer type was also performed based on four separated population groups: mixed, European, African, and Asian. For data to be considered significant, *p* ≤ 0.05 and hazard ratio (HR) < 1 (increased survival associated with higher *COIL* expression) or >1 (reduced survival associated with higher *COIL* expression).

### 2.11. Statistical Analysis

Statistical analysis was performed using the GraphPad Prism^®^ 6 software (v6.02, GraphPad Software Inc., San Diego, CA, USA). Error bars in the bar charts represent mean ± standard deviation of at least three independent experiments. Two-tailed Student’s *t*-test was used to determine significant differences between the means of analyzed data. *p* ≤ 0.05 was considered to be statistically significant.

## 3. Results

### 3.1. Global Distribution of the Coilin Nonsynonymous SNPs (rs116022828 and rs61731978)

We searched dbSNP (https://www.ncbi.nlm.nih.gov/snp/) and analyzed the allele frequencies of both SNPs in different populations to explore the global distributions of rs116022828 (E121K) and rs61731978 (V145I). By comparing the frequency data across multiple datasets, we found that rs116022828 was predominantly occurred in the African population, with a frequency greater than 0.03 (Appendix A). As for rs61731978, it mainly occurred in the American population with an extremely high frequency of >0.2 (Appendix A). In addition to the American population, rs61731978 also appeared in substantial numbers of African, Asian (South Asian), and European populations with frequencies ≥0.01 (Appendix A). The above results indicated that rs61731978 is more globally distributed, whereas rs116022828 is more restricted to the African population.

### 3.2. Establishment of HeLa Cell Lines Expressing Coilin WT or SNP Variants

CBs are more abundant in transcriptionally active cancer cells [31]. Here, we compared the numbers of CBs by observing coilin foci in a normal human lung cell line (BEAS-2B) against two lung cancer cell lines (A549 and H1299) and a cervical cancer cell line (HeLa). Based on the observation by immunofluorescence microscopy, cells from all three cancer cell lines were indeed harbored significantly more CBs than the normal BEAS-2B cells (Appendix A).

The HeLa cells are typically used for coilin research due to their higher transcriptional activity [32,33]. The *COIL* gene in the HeLa cells was sequenced to check for mutations, and the results indicated that the cells contained coilin WT (no mutation and SNP). Thus, in order to assess the functional impacts of the two coilin SNPs (E121K and V145I, Figure 1a) on CB formation and other phenotypes, we transiently or stably transfected HeLa cells with either the pEGFP-C1-Coilin-WT (as control), pEGFP-C1-Coilin-E121K, or pEGFP-C1-Coilin-V145I plasmid (Figure 1b). Subsequently, the CB formations between the cell lines generated by transient transfection and stable transfection were compared, and the cell lines with the higher CB formation rate were selected for further experiments. Figure 1c summarizes the experimental design and workflow of this study.

### 3.3. Effects of Transiently and Stably Transfected Coilin SNP Variant Cells on CB Formation

HeLa cells that were transfected with coilin WT or SNP variants were examined using a fluorescence microscope and observed for CB formation. According to previous research, high levels of coilin expression, usually associated with transient transfection, could disrupt coilin localization and CB formation, resulting in the dispersion of coilin in the nucleoplasm [10]. Thus, two forms of coilin expression were anticipated in the transfected cells, as demonstrated in Figure 2a: cells with normal CB formation (coilin localized to the CBs resulting in distinct EGFP-coilin foci) and cells with abnormal overexpressed coilin (dispersion of coilin throughout the nucleoplasm). Indeed, the results from the immunofluorescence analysis indicated that for transiently transfected cell lines, only less than 50% of WT cells and less than 30% of E121K or V145I cells (*n* > 100) displayed normal CB formation (Figure 2b). In contrast, for stably transfected cell lines, more than 89% of the WT, E121K, or V145I cells (*n* > 140) showed normal CB formation (Figure 2c). Based on these findings, we selected the stably transfected cell lines to perform the following experiments in this study.

### 3.4. Effects of Coilin SNP Variants on Subcellular Co-Localization with SMN

Coilin and SMN are the two main components among the identified CB proteome [9,34,35]. We performed immunofluorescence analysis on the stably transfected cell lines to detect their coilin and SMN co-localization status in order to determine whether the coilin SNP variants have impacts on the subcellular co-localization of coilin and SMN. Overall, it was revealed that the co-localization of coilin and SMN in these cell lines showed a range of patterns (Figure 3a; data of positive/negative controls is shown in Appendix A).

Here, we found that the WT cells contained considerably more “canonical CBs” (the CBs that contained both coilin and SMN; indicated by single arrow) as compared to the coilin SNP variant cells. Unlike the WT cells, many of the E121K and V145I cells contained coilin foci, but without SMN enrichment (indicated by double arrow). These coilin foci lacking SMN protein are termed as “residual CBs”, and they are known to be associated with abnormal function of CBs [36]. Interestingly, we also observed different patterns of coilin and SMN co-localization that were mainly restricted only to a particular coilin SNP variant cell line. For instance, in many of the E121K cells, the coilin proteins appeared to be accumulated in the nucleoli (indicated by single arrowhead).

The co-localization status of coilin and SMN between the cell lines was summarized based on three groupings in Figure 3b: cells that contained only the canonical CBs (“completely”), cells with mixed co-localization status (“partially”), and cells with no coilin and SMN co-localization (“none”). The results indicated that about 11–14% of the cells from all three cell lines contained no coilin and SMN co-localization, and it is clear that the WT cells contained a noticeably greater number of canonical CBs than the two coilin SNP variant cells (75.27% vs. 25.87–42.45%) (Figure 3b).

### 3.5. Effects of Coilin SNP Variants on Microtubule Formation, Cell Cycle, and Cell Proliferation

We then investigated whether coilin SNP variants have effects on the growth-related phenotypes of the HeLa cells. We performed microtubule regrowth assay to assess the microtubule formation in the coilin WT and SNP variant cells. The results indicated that 1 h after the removal of nocodazole, multiple nuclei were observed in the E121K cells compared with WT and V145I cells (Figure 4a). These findings suggested that more of the E121K cells might be trapped in the S and G2/M phases of cell cycle compared to the WT and V145I cells. Subsequent cell cycle analysis on these cell lines proved that our assumption was accurate (Figure 4b). In cell proliferation assays, we found that the V145I cells had a similar growth rate with the WT cells, and the E121K cells were growing slower than the WT cells, as expected (Figure 4c).

### 3.6. Effects of Coilin SNP Variants on Coilin Expression

We performed immunoblotting to examine the coilin protein expression level (Figure 5a; Appendix A) and RT-qPCR for the detection of *COIL* gene expression levels (Figure 5b) in order to elucidate whether coilin SNP variants could affect the expression of coilin. The results from the immunoblotting and RT-qPCR were generally in agreement with each other. Notably, the WT and E121K cells showed similar lower coilin expression levels, and the V145I cells had the highest coilin expression.

### 3.7. In Silico Three-Dimensional Modeling and Bioinformatic Analysis of Coilin WT and SNP Variants

There is currently no crystal structure of coilin available. In this study, the AA sequences of coilin WT and two SNP variants were sent to the I-TASSER server for three-dimensional protein modeling analysis [28]. The predicted normalized B-factors indicated that the two ends (the first ~100 AA and last ~100 AA) of the coilin protein appeared to be more stable than the middle section (Appendix A). Additionally, data also indicated that the E121K mutation resulted in the disappearances of the helix structure in the E121K mutant (Appendix A, indicated by red arrow).

We then used STRUM to predict the stability of coilin upon a single-point mutation [29]. The results indicated that the E121K mutation had a ΔΔG value of −2.93 while V145I was 0.18 (Figure 6a). A positive ΔΔG value indicates that the mutation induces protein stability, whereas a negative value causes destabilization.

In regard to protein structure modeling, five models with the highest C-scores were generated for each of the AA sequences submitted to I-TASSER (Appendix A). Based on the models with the highest C-scores, it appeared that the E121K mutation had resulted in the most dramatic change in the predicted coilin structure (Figure 6b).

### 3.8. Clinical Bioinformatic Analysis: Relationships Between Coilin Expression and Cancer Survival

We screened through The Cancer Genome Atlas (TCGA) database searching for cancer patients with coilin mutation at residue 121 or 145 to further investigate the relationships between coilin and cancer [37]. The data indicated that these coilin mutations were extremely rare in cancer patients, suggesting that coilin SNP variants are likely not cancer risk factors (data not shown). However, since our previous data indicated that coilin SNP variants could give rise to different levels of protein and gene expressions, we then sought to determine the roles of coilin expression in cancer survival. Accordingly, we used Kaplan–Meier Plotter database [30] to perform survival analysis on cancer patients with high/low *COIL* expression across 21 cancer types (Appendix A). The results showed that high *COIL* expressions were significantly correlated with lower (HR > 1) or higher (HR < 1) survival outcomes in specific cancer types, as shown in Figure 7a.

We then used the same datasets to analyze the survival rates based on three separate populations: European, African, and Asian (Figure 7b–d). We found that the survival patterns in the European population were virtually the same as the mixed population analysis, with the exception of lung squamous cell carcinoma, where it only showed up in the European population (Figure 7b). For African and Asian, the survival analyses for some of the cancers were invalid due to the low number of patient samples (labeled as N/A). When focused only on the African datasets, survivals of four cancer types (bladder carcinoma, cervical squamous cell carcinoma, sarcoma, and uterine corpus endometrial carcinoma) appeared to be positively or negatively correlated with the *COIL* expressions (Figure 7c). Among them, bladder carcinoma, cervical squamous cell carcinoma, and sarcoma showed significant correlations with *COIL* expressions only in the African population, but not in the mixed and other populations (Figure 7c). In the Asian population, survivals of three cancer types (head-neck squamous cell carcinoma, liver hepatocellular carcinoma, and stomach adenocarcinoma) were significantly correlated with *COIL* expressions, and stomach adenocarcinoma was only found to be related to *COIL* expression in this population (Figure 7d).

Detailed data of clinical bioinformatic analysis regarding coilin expression and cancer survival are summarized in Figure 7e. In most cases, the significant correlations between *COIL* expression and survival of a particular cancer type, for it be a positive or negative correlation, are consistent across different populations (Figure 7e). However, for head-neck squamous cell carcinoma, high *COIL* expressions are correlated with greater survival in the European but lower survival in the Asian (Figure 7e). Overall, the above results suggest that coilin expression may play different roles in cancers, depending on the cancer types as well as populations.

## 4. Discussion

The CB was first described as a nucleolar accessory body by Santiago Ramón y Cajal in the 1900s, and later confirmed by electron microscopy [38]. Over the years, research on this coiled-like structure has revealed that CBs are involved in the modification, assembly, and trafficking of snRNPs and snoRNPs, as well as the assembly and regulation of telomerase [1,31]. Studies have also identified a range of proteins that localized to the CBs and, among them, coilin is widely recognized as the main component of, and a molecular marker for, CBs [2,5,39].

There are several examples of human diseases that are highly associated with the disruption and loss of CBs, and they usually arise due to mutations in the genes that encode CB-associated proteins. For instance, SMA, where patients are characterized by progressive degeneration of spinal motor neurons, muscle atrophy, symmetric limb paralysis, and respiratory failure, is caused by mutated *SMN1* gene [14,40]. SMN is considered to be one of the most important components of a canonical CB in addition to coilin [41]. Another genetic disease, known as dyskeratosis congenita (DKC), occurs due to mutations in a couple of genes encoding proteins that localize to CBs (e.g., *EST1A*, *NHP2*, *NOP10*, and *WRAP53*), and resulting in defective ribosome biogenesis and telomere maintenance [15,42]. DKC is a rare X-linked recessive disorder that is characterized by bone marrow failure, premature aging, skin and appendage lesions, and predisposition of cancer [15]. However, it is noteworthy that not all diseases of ribosome biogenesis are CB-related (reviewed in [43]), and conditions, such as the dosage of ribosomal genes, could result in abnormal ribosome biogenesis [44].

Apparently, CBs also play a role in tumorigenesis. Many studies, including ours (Appendix A), have shown that CBs are enriched in the cancer cells, probably owing to their higher transcriptional activity [2,17]. Surprisingly, although there is enough evidence to indicate the connections between CBs and specific human pathologies, the roles of coilin in pathogenesis and tumorigenesis have remained largely unknown.

Previous studies have shown that modifications in coilin residues can impact CB formation [32,33]. However, the effects of the two nonsynonymous SNPs of *COIL* gene (rs116022828:E121K and rs61731978:V145I) on CB formation and other cell phenotypes have not been investigated. Here, we constructed HeLa cell lines with coilin WT or coilin SNP variant (E121K or V145I) and investigated, for the first time, how coilin SNPs affect CB formation, co-localization with SMN, cell growth, and coilin expression and structure.

We observed a substantial number of cells from the transiently transfected cell lines contained abnormal overexpressed coilin that dispersed throughout the nucleoplasm. This phenomenon has also been observed in, and is in agreement with, other studies, indicating that overexpressed coilin could disrupt CB formation [10]. In addition to coilin, the overexpression of a CB-associated protein WRAP53 has also been reported to disrupt CB [15]. These data altogether suggest that a stable cell line that expressed moderate levels of CB-associated proteins is more suitable to be used in CB research, since CBs are likely to be disrupted when high levels of CB-associated proteins are exogenously expressed.

Coilin and SMN are considered the two most important components in CBs [9,34]. When comparing the subcellular co-localization of coilin and SMN in the coilin WT and SNP variants, we found that both SNP variant expressing cell lines had lesser cells with canonical CBs (normal CBs that contained both coilin and SMN), but many with residual CBs (coilin foci without SMN). Interestingly, it appeared that the cells with a mutation at coilin residue 121 contained a more diverse pattern of coilin and SMN co-localization. For example, accumulation of coilin in the nucleoli and/or gem structure formation (SMN foci) could frequently be found in the E121K cells but rarely in the WT and V145I cells. The E121K coilin might be targeted to nucleoli by nucleolar and coiled-body phosphoprotein 1 (NOLC1) since it was previously demonstrated that the N-terminal of coilin interacted strongly with NOLC1 [45]. Nevertheless, the findings in our study not only show that coilin SNP variants have impacts on the CB formation and coilin/SMN co-localization, but also indicate that the consequences of E121K and V145I mutations are different. Furthermore, it is indicated that the coilin E121K mutation has greater impacts on the CB formation than the V145I, which warrants further study.

Some AA residues can undergo post-translational modifications (PTMs). For instance, it was reported that the glutamic acid (E) has the potential to be methylated [46]. Lysine (K), on the other hand, is subjected to a wide range of PTMs, including acetylation, methylation, phosphorylation, ubiquitylation, and more [47]. However, it remains unclear whether the residue at position 121 of coilin (E/K) could subject to these PTMs, but previous research on other residues of coilin have indicated that PTMs in coilin are important for CB formation. Specifically, it was previously shown that coilin localization was regulated by its methylation state [17]. The co-localization of coilin with SMN in the CBs also appeared to be controlled by methylation [27,48,49]. Moreover, it was shown that a high percentage of HeLa cells stably expressing S489D-mutated coilin (mimic phosphorylation) had their coilin accumulated in the nucleoli, similar to our E121K cells [50].

CBs are involved in cell development and proliferation [6,7,8]. HeLa cells with impaired CBs or knocked-down coilin expression showed reduced cell proliferation [50,51]. In animal models, coilin knockout mice displayed significant fertility and fecundity defects [52]. It was also showed that the knockdown of coilin caused embryonic lethality in the zebrafish, indicating the essentiality of coilin in embryogenesis [53]. In the current study, we discovered that coilin E121K variant cells had significant differences in their microtubule formation and cell cycle when compared to coilin WT and V145I cells. Both microtubule regrowth assay and cell cycle assay suggested that a large proportion of E121K cells might be trapped in the S and G2/M phases of cell cycle, and was likely to be the cause of reduced cell proliferation. Coincide with the CB formation and coilin/SMN co-localization results, the cell proliferation results indicate that the mutation of coilin residue 121 has greater impacts on the cell growth than residue 145. However, it is important to mention that our study has limitations since the effects of coilin SNPs may be compensated by the expression of endogenous coilin. Additional study is required in order to verify our findings using cell lines with knocked off/down coilin or preferably, CRISPR-generated cell lines expressing the coilin SNP mutants.

For the season that there is currently no crystal structure of coilin protein, we performed a three-dimensional protein modeling analysis for coilin WT, along with the two coilin SNP variants, while using I-TASSER server [28]. The results indicated that the coilin with E121K mutation appeared to have a distinctively different predicted protein structure than the others based on the highest C-scores. Additionally, predicted normalized B-factors indicated that a large portion of the coilin structure, especially in the middle section, was relatively unstable, and we believed that this might make it harder to predict the structure of coilin. Furthermore, we used STRUM to predict the effects of E121K or V145I mutation on the coilin stability through assessing the thermodynamics of the protein [29]. The results indicated that the E121K mutation caused destabilization on the coilin protein structure, whereas V145I mutation slightly stabilized the protein.

The connections between coilin and cancer have not been systematically assessed. In recent years, the associations of SNPs and disease risks have been demonstrated in many studies [22,23,24]. Here, we performed clinical bioinformatic analysis and found that coilin mutation at residue 121 or 145 is rare in patients of various cancer types. However, subsequent analysis showed that the coilin expression levels correlated with patient survival in some cancer types, and that population might also be a factor. Thus, these results indicated that coilin SNP variants are likely not cancer risk factors, but the expression of coilin could be. Previous studies have indicated that SNP can alter protein structure and cause various diseases, but it is so far unclear how SNP can affect gene and protein expression. Interestingly, here we showed that coilin SNP variants could potentially give rise to different levels of protein and gene expressions. We hypothesize that certain point mutations on exon can potentially change the secondary structure of mRNA and, hence, affect the efficiency of gene transcription and translation, but it would require substantial further work to verify this. Overall, findings from our clinical bioinformatic analysis may have potential clinical implications on cancer patients of different populations: the E121K SNP is primarily distributed in the African population, whereas the V145I SNP is mainly distributed in the American population but also observed in African, South Asian, and European populations [54].

## 5. Conclusions

In conclusion, we demonstrated, for the first time, that coilin SNP variants, E121K and V145I, could differentially impact CB formation, co-localization with SMN, microtubule formation, cell cycle, cell proliferation, and coilin expression. Among the two SNP variants, the E121K variant appeared to have greater detrimental effects on the cells than the V145I variant. Furthermore, clinical bioinformatic analysis indicates that coilin expression may potentially be a risk factor for cancer, depending on the cancer types and races.

## Figures and Tables

**Figure 1 genes-11-00895-f001:**
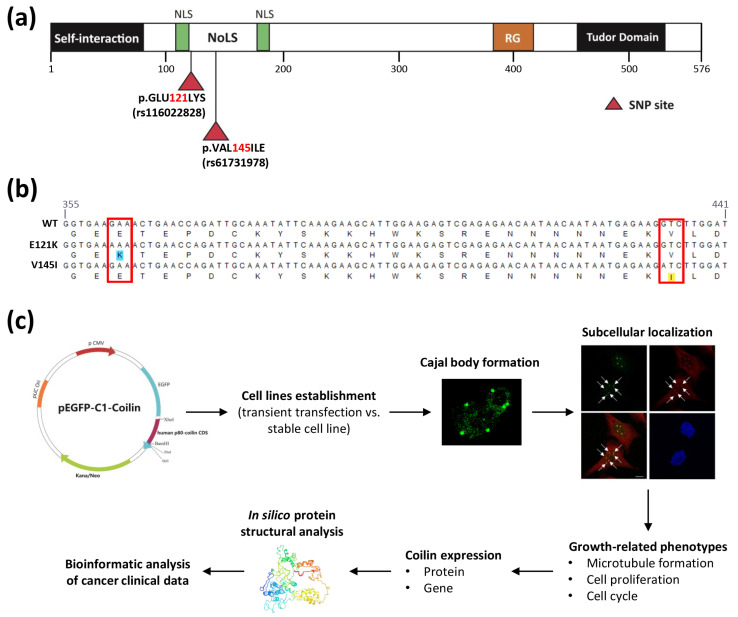
Establishment of cervical cancer (HeLa) cell lines expressing coilin single nucleotide polymorphism (SNP) variants. (**a**) Schematic structure of human coilin containing the self-interaction domain, nuclear localization signals (NLSs), nucleolar localization signal (NoLS), tri-RG motif, and tudor domain. The two coilin nonsynonymous SNPs (rs116022828 and rs61731978) are located between the NLSs where a cryptic NoLS is positioned. (**b**) DNA and amino acid sequence alignments of coilin SNP sites. For rs116022828, there is a Glu-to-Lys substitution at position 121 (E121K), while for rs61731978, there is a Val-to-Ile substitution at position 145 (V145I). (**c**) Experimental design and workflow in this study.

**Figure 2 genes-11-00895-f002:**
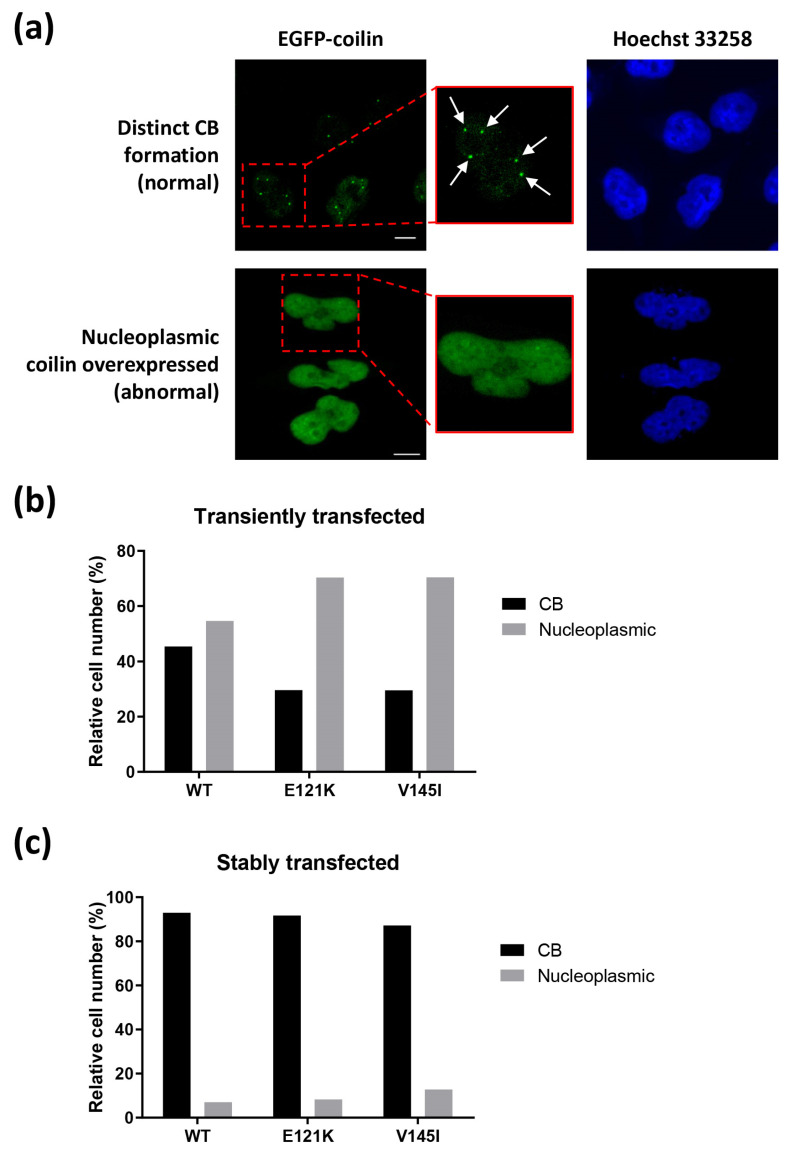
Effects of transiently and stably expressed coilin wild-type (WT) and SNP variants on Cajal body (CB) formation. (**a**) Immunofluorescence analysis of HeLa cells with normal (indicated by single arrow) and abnormal CB formation; scale bar = 10 µm. (**b**) HeLa cells were transiently transfected with coilin WT (*n* = 108), E121K (*n* = 118), or V145I (*n* = 132) and observed for CB formation. (**c**) HeLa cells were stably transfected with coilin WT (n = 156), E121K (*n* = 145), or V145I (*n* = 180) and observed for CB formation.

**Figure 3 genes-11-00895-f003:**
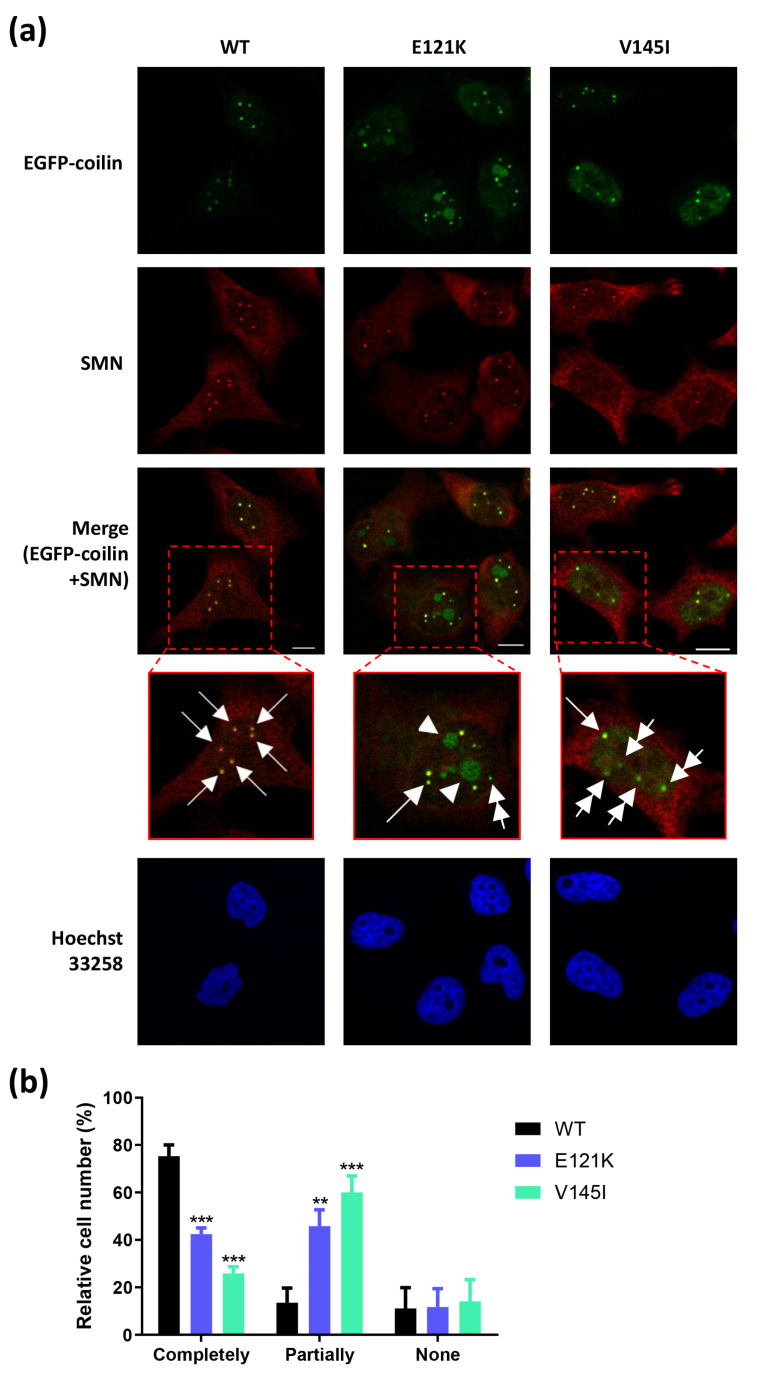
The effects of coilin WT and SNP variants on subcellular co-localization of coilin and SMN. (**a**) Immunofluorescence analysis of coilin and SMN co-localization status; single arrow indicates typical CB, which consists of the coilin and SMN protein; double arrow indicates residual CB, referring to the coilin lacking SMN protein; single arrowhead indicates coilin accumulation in the nucleolus; scale bar = 10 µm. (**b**) Co-localization status of coilin and SMN in WT (*n* = 95–157), E121K (*n* = 114–161), and V145I (*n* = 127–166) cells were separated into three groupings: cells contained only the canonical CBs (completely), cells with mixed co-localization status (partially), and cells with no coilin and SMN co-localization (none); Student’s *t*-test was performed to determine significant differences between the means of SNP variant cells and control WT cells. ** *p* ≤ 0.01; *** *p* ≤ 0.001.

**Figure 4 genes-11-00895-f004:**
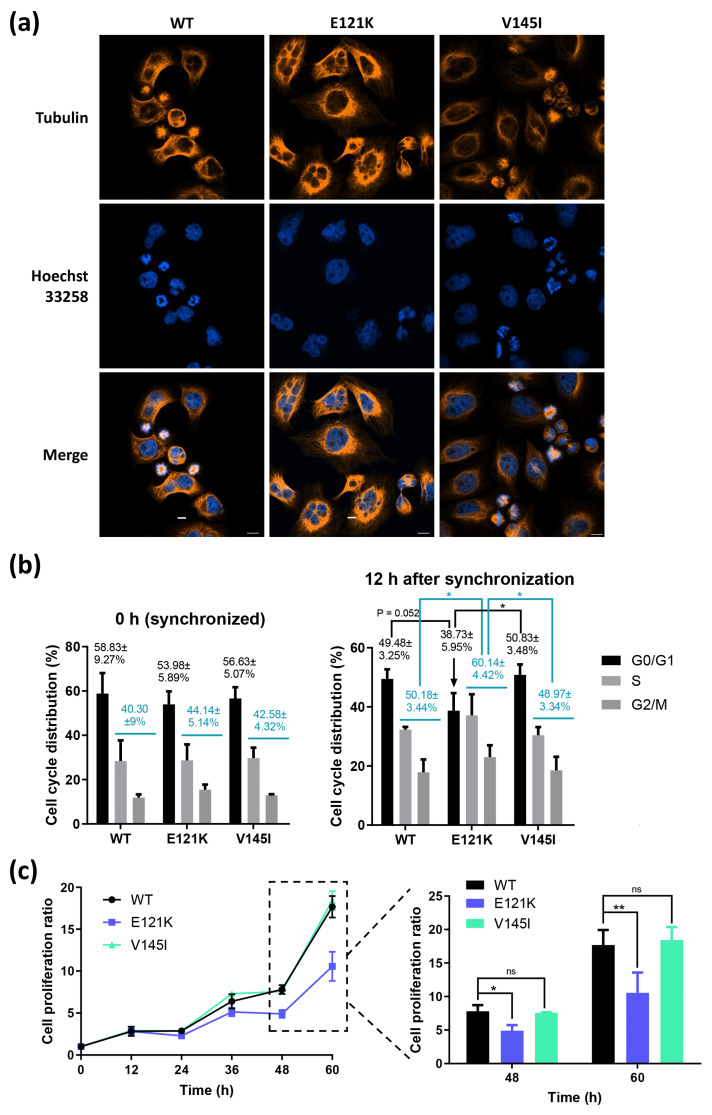
Effects of coilin SNP variants on cell growth-related phenotypes. (**a**) Microtubule regrowth assay: coilin WT and SNP variant cells were treated with nocodazole for 16 h, and stained with tubulin antibody after 1 h to visualize microtubules upon regrowth; scale bar = 10 µm (**b**) Cell cycle distribution was measured by the flow cytometry: cells were starved in FBS-free culture medium for 24 h and stimulated with 10% FBS for 12 h. Numbers highlighted in blue represent the % cell distributions of S phase + G2/M phase. (**c**) Cell proliferation was determined by MTS assay in a 12 h interval for up to 60 h. * *p* ≤ 0.05; ** *p* ≤ 0.01; ns: not significant.

**Figure 5 genes-11-00895-f005:**
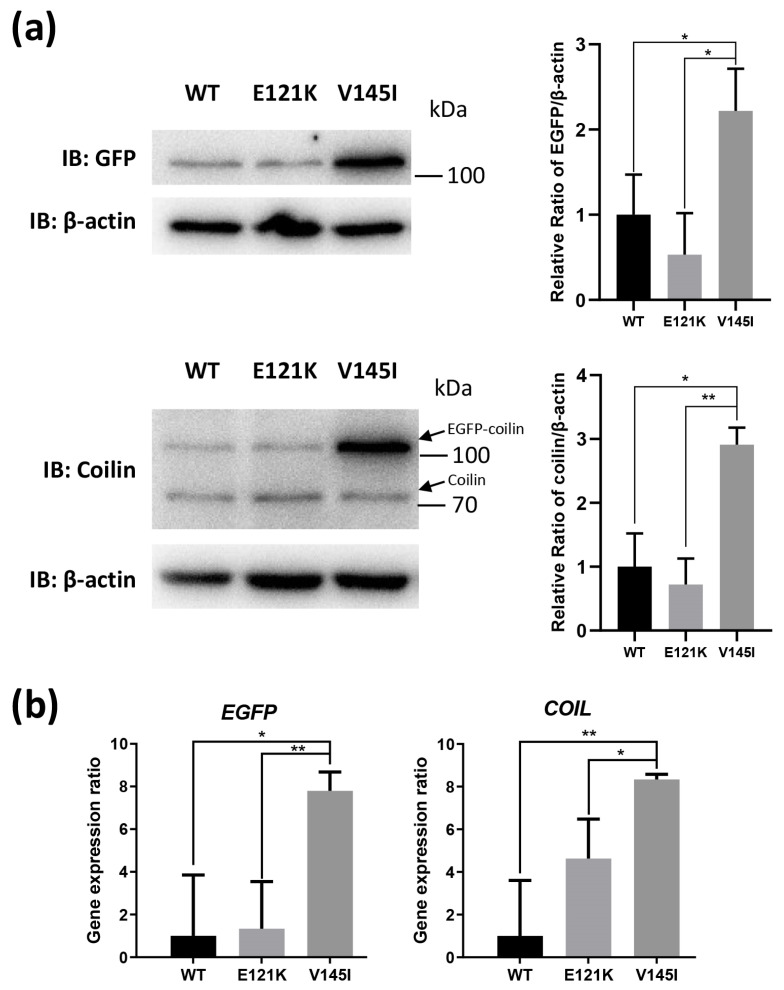
Effects of coilin SNP variants on coilin protein and gene expression. (**a**) Immunoblotting of coilin WT and SNP variant cells probed with GFP (sc-9996; EGFP-coilin ≈ 110 kDa) or coilin (sc-55594; coilin ≈ 80 kDa) antibody; β-actin (A5441; ≈42 kDa) was served as an internal loading control (see Appendix A for full blot). Figures are representative of three independent experiments with similar expression trends. (**b**) Gene expression of coilin in the three cell lines was analyzed using RT-qPCR; the *ATCB* gene was used as an internal reference to normalize gene expression. * *p* ≤ 0.05; ** *p* ≤ 0.01.

**Figure 6 genes-11-00895-f006:**
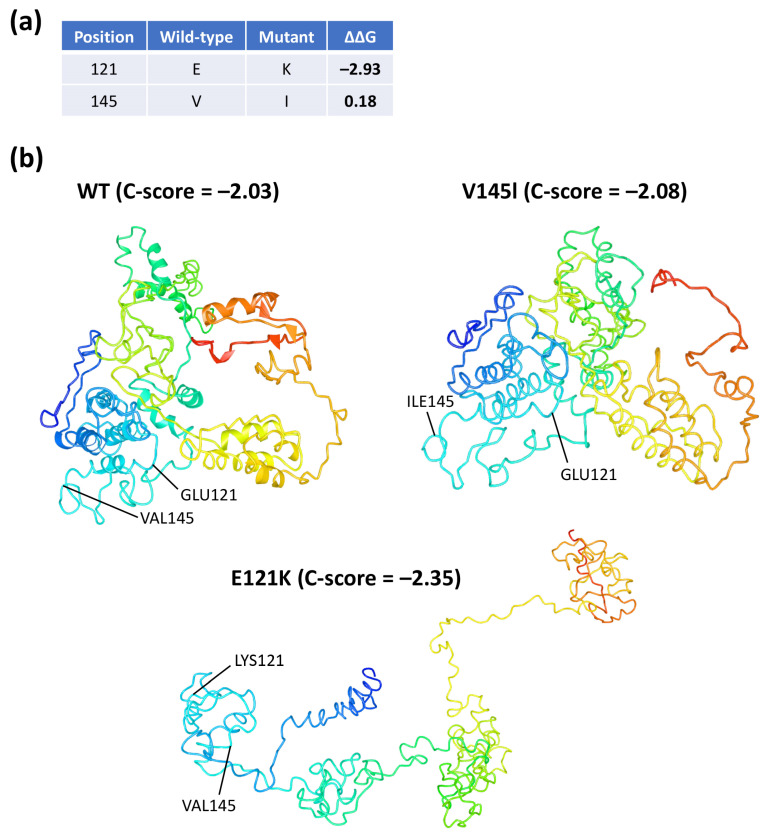
Coilin in silico structural prediction and analysis. (**a**) Protein stability was predicted by STRUM (https://zhanglab.ccmb.med.umich.edu/STRUM). A positive ΔΔG value indicates the mutation induces protein stability whereas a negative value causes destabilization [29]. (**b**) Protein models with the highest C-scores predicted by I-TASSER server (https://zhanglab.ccmb.med.umich.edu/ITASSER) using the amino acid sequences of the coilin WT and SNP variants [28]. The displayed proteins were modified and generated using NCBI’s iCn3D.

**Figure 7 genes-11-00895-f007:**
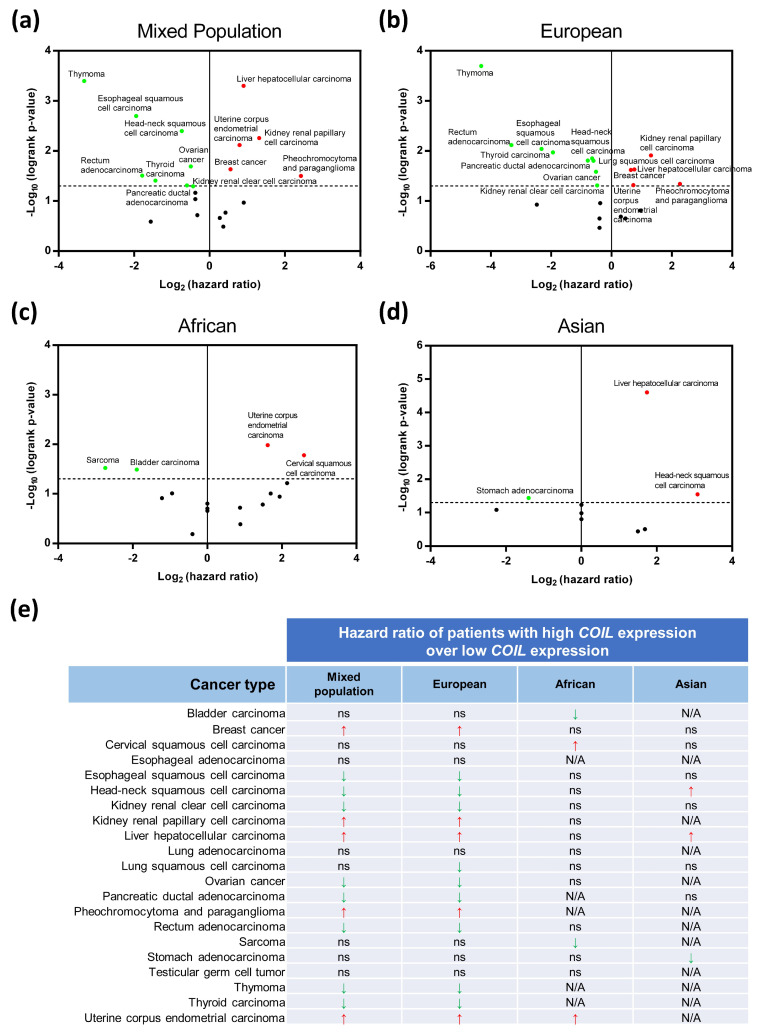
Survival analysis of cancer patients with high versus low coilin gene expressions using Kaplan–Meier Plotter database (https://kmplot.com/analysis). Analyses were carried out in 21 types of cancers, and based on four different population groups, (**a**) mixed population, (**b**) European, (**c**) African, and (**d**) Asian. In panels a–d, the dash line on the y-axis represents *p* ≤ 0.05; the vertical line on the x-axis represents hazard ratio (HR) at 0; (**e**) Summary of the bioinformatic analysis results in panels **a–d**; ns = non-significant; N/A = not applicable (due to no data or low in sample size).

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
