# Peer review of "The Impact of Coilin Nonsynonymous SNP Variants E121K and V145I on Cell Growth and Cajal Body Formation: The First Characterization"

_genes, 2020, doi:10.3390/genes11080895_

Round 1

Reviewer 1 Report

I would like to thank the authors for addressing my concerns. In my opinion, the revised manuscript is suitable for publication in Genes.

There are minor points for improvements:

The authors should add small insets in the corners of microscopic figures showing highlighted Cajal bodies by arrows with much higher magnification. They are poorly visible now.

Line 46: “it is considered as the central element of the CB” … change it to “as the architectural element and marker protein of the CB”

Line 209: change the sentence to “Coilin and SMN are the two main components in CBs [9,28]” among the identified CB proteome (doi: 10.1002/wrna.1514).

Line 220: “in many of the E121K cells the coilin proteins appeared to be accumulated in the nucleoli” The authors should hypothesize in Discussion that E121K coilin is likely targeted to nucleoli by the low complexity central region of Nopp140/NOLC1, which is a strong interactor with the N-terminal portion of coilin (doi: 10.1083/jcb.142.2.319).

Reviewer 2 Report

The manuscript consists of two parts: first, an experimental study of coilin SNP effects of coilin transcription, CB patterns, cell proliferation and microtube formation, and second, a bioinformatic assays on correlation between the SNP and cancer, and the possible structure of coilin proteins with different SNP.

The second part in beyond my area of expertise, and below I'd like to draw the authors' attention to the errors and questions predominately within the first part of the study.

My comments below are given in the order they stay in the manuscript.

Line 27: 'contained ... accumulation' - better 'showed [demonstrated] ... accumulation' or 'contained ... aggregates [granules, precipitates]"

Line 40: '...human, animal, plant, yeast, amphibian, and insect'. As amphibian, and insect are animal, it seems that 'animal' is redundant. Did the authors mean 'mammalian'?

Line 56-57: 'autosomal recessive... disease ... with high carrier frequency - about one in 30 to 40". The SMA genetics is more complicated than a single gene model implicated in this short description. The real prevalence is about 1 to 6000, and this should be noted in the introductuion.

Line 74: 'the roles of coilin nonsynonymous SNPs on CB formation...' - the preposition 'on' is wrong after 'role'. 'In' should be used instead, or 'effect...on' is also correct.

Lines 222-227 and 234-235 and 247-253: the authors seem to compare the percentages (frequencies) using Student's t-test. But t-test is for comparing means of samples of continuous values. When the percentages are to be compared, the statistics for qualitative variables must be used, such as gender, colour, belonging to a group. They are chi-square or Fisher's exact test.

Line 254 and further (clause 3.6). It's unclear, how SNP can affect transcription and translation? Typically, SNP alter the protein structure, but not expression. So, this part should be reconsidered.

Clauses 3.7 and 3.8: I'm not an expert in this field, so I can ask common questions only:
- did you use the Bonferroni correction for multiple comparison?
- can the correlation between SNP and cancer types caused by different cancer occurence in different populations?

Line 341:  you mentioned a CB-related disease (diskeratosis cogenita) resulting in defective ribosome biogenesis. Some more examples of  other (though CB-unrelated) diseases of ribosome biogenesis, such as https://www.ncbi.nlm.nih.gov/pmc/articles/PMC2965583/, and other conditions with ribosome involment (e.g. https://pubmed.ncbi.nlm.nih.gov/30343462/ ) would improve the discussion.

Line 366: 'colin' should be corrected with 'coilin'.

Line 397: [23] (Roy...) - the reference (Roy, Kucukural, & Zhang, 2010) should be deleted, as this is the same as [23]

Summarizing, the manuscript can be accepted with minor corrections.

Reviewer 3 Report

The authors explore the impact of nonsynonymous SNP variants of coilin protein on cell growth and cajal body formation. The topic is important for understanding of cancer mechanisms and has broad interest to the audience. The writing of the manuscript is easy to follow, however, it also has major problems to address, see below.

1, The plasmid construct uses a CMV promoter to drive the expression of mutant coilin with a EGFP reporter. Due to the feature of exogenous promoter, the stable cell line isolated can still have variant expression levels from time to time and from clone to clone. The different level of coilin-EGFP expression shown in Figure 5 may not come from the mutant but simply the variance of exogenous expression. Question: what is the SD of coilin-EGFP expression from Western blot analysis from experimental repeats? The author should attempt to select clones with equivalent levels of coilin-EGFP expression from both mutants and compare the phenotypic impacts. Otherwise, if the mutant leads to different level of protein expression, the author should provide adequate WB quantification and statistical analysis. 

2, Another major concern related with the above comment is, the mutant effect may be compensated by the expression of the natural gene copy. It is hard to know to what extent of the phenotype is coming from the mutant.

3, Figure 2, how was the quantification carried out? The author should specify the categorizing strategy for CB and nucleoplasmic cells. What does the n mean?

4, Figure 3, image presentation needs to be improved, especially for the SMN staining, looking very faint. The authors also need to provide negative controls to validate the real positive signal. How was (b) quantified? What is the analysis method?

5, Figure 4a, can not see how "less cells undergoing mitosis"

In terms of writing, the authors need to include references to a lot of the statements in introduction and other parts. For example, line 62,63,64, 68,69,73,158,166

Minor point: line 99, what is the lens used for imaging? 

Round 2

Reviewer 3 Report

The authors did a good job on addressing the concerns raised. The endogenous level of coilin expression is still the major concern and the author's response about endogenous expression being interrupted by exogenous expression is not solidly supported. The systematic investigation into this may bring new important findings in the future.